# Echo: A multi-agent AI system for patient-centered pharmacovigilance

**Claude Sonnet 3.5**
Anthropic

**Megha Srivastava**
Stanford University
megha@cs.stanford.edu

## Abstract

Online health communities provide patients with spaces to share experiences, find support, and voice concerns that may go unacknowledged in clinical encounters. These narratives often include early reports of adverse drug reactions (ADRs), yet remain largely absent from formal pharmacovigilance. We present Echo, a multi-agent AI system that transforms patient narratives from Reddit into structured drug safety intelligence. Echo deploys four specialized language model agents in concert: an Explorer mining social media forums, an Analyzer quantifying associations through temporal, confidence, and community metrics, a Verifier identifying novel signals absent from FDA databases, and a Proposer generating testable hypotheses from biomedical literature. As a proof-of-concept, we show that from less than 200 Reddit posts, Echo was able to discover 640 drug-symptom associations, including several absent from official databases, such as pembrolizumab-induced daytime somnolence. We further show in retrospective case studies that Echo might have detected emerging toxicities, such as checkpoint inhibitor pneumonitis, before regulatory recognition. Beyond signal detection, Echo also identifies confounding factors and proposes testable hypotheses. Finally, we build an interactive interface to help explore associations, examine patient quotes, and access AI-generated insights. Overall, Echo leverages language models to surface patient-reported signals that may complement regulatory surveillance.

## 1 Introduction

Pharmacovigilance—the science of monitoring and improving drug safety in the real world—is both a clinical necessity and a human imperative. Clinical trials remain the gold standard for demonstrating efficacy, but they are not designed to capture the full distribution of rare, long-term, or population-specific adverse drug reactions (ADRs) that only emerge once a therapy is broadly deployed [Arrowsmith, 2011, Hazell and Shakir, 2006]. Traditional post-marketing surveillance relies heavily on clinician-initiated reporting (e.g., to the FDA Adverse Event Reporting System), which is often delayed, incomplete, and shaped by systemic underreporting. As a result, dangerous toxicities may circulate silently for months or years before receiving formal recognition. The stakes are highest in oncology, where new therapies like checkpoint inhibitors and kinase inhibitors are rapidly changing survival curves but can unleash immune-related or metabolic toxicities with potentially life-threatening consequences.

Meanwhile, patients themselves are narrating their therapeutic journeys in unprecedented detail online. On Reddit and other forums, individuals living with cancer build communities where they share treatment decisions, side effects, and coping strategies—not as curated clinical records, but as authentic human testimony [Sarker and Gonzalez, 2015, Lavertu et al., 2021]. These conversations contain signals that traditional pharmacovigilance cannot access: symptoms reported outside of clinic visits, rare or embarrassing experiences that patients might not disclose to physicians, or early whispers of adverse events before they are recognized formally (see Figure 1). Harnessing such

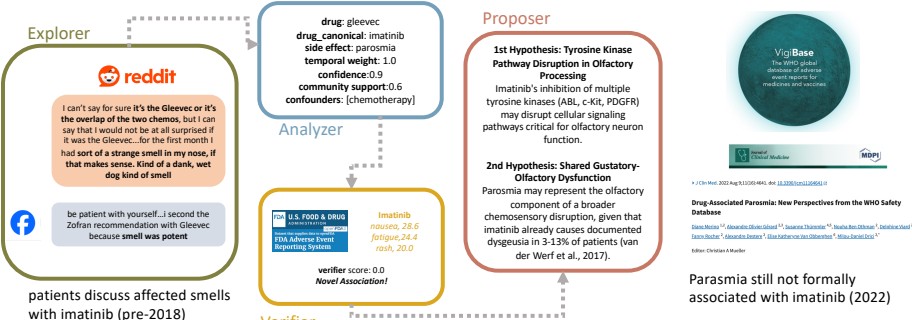

Figure 1: Overview of **Echo** . Online patient discussion of smell distortions while tasking Gleevec (imatinib) is discovered and then compared against official databases and proposed as a novel adverse drug reaction, with a potential confounder of concurrent chemotherapy noted.

signals requires care but offers an extraordinary opportunity: to hear the patient's voice in real time and augment existing safety systems.

Prior work provided strong evidence that patient posts from the Inspire community health forum can help identify overlooked adverse drug reactions, focusing specifically on skin-related ADRs to chemotherapy drugs [Ransohoff et al., 2018]. However, not only was this study retrospective (the research team knew to look for cutaneous ADRs), but it also required researchers to manually label patient posts in order to train DeepHealthMiner, a neural-network-based named entity recognition (NER) system, to detect the different ways patients expressed adverse reactions (e.g. *"my arms are itchy"*, *"I keep wanting to scratch my skin!"*) [Nikfarjam, 2016]. Thus, their approach cannot scale to discover the broad set of (drug, side effect) reactions there exists in oncology, nor the rich variety of ways patients express their drug experience. Furthermore, DeepHealthMiner is unable to analyze the associations for important factors such as the temporal relationship between drug usage and side effects, the patient's confidence in the adverse effect, support from other members of the community forum, and potential confounding variables for the association (e.g. a patient's allergy to a new pet causing a skin reaction). Finally, Ransohoff et al. [2018] manually compare identified associations with the medical literature to demonstrate novelty, rather than systematically verify which signals are missing or underreported.

To address these limitations, we create **Echo** , a multi-agent AI system that amplifies patient voices from public social media forums to discover, validate, and contextualize novel drug-symptom associations. **Echo** takes advantage of recent advances in large language models to create a more systematic, quantitative, and forward-looking pharmacovigilance platform than prior work [Ransohoff et al., 2018], and is designed as a collaborative team of specialized agents, each with a well-defined role:

- **Explorer** : Systematically extracts drug-symptom mentions from cancer subreddit discussions (e.g. `/r/cancer`), handling the informal language of patient narratives.
- **Analyzer** : Quantifies association strength through three metrics—temporal proximity, patient confidence, and community engagement—enabling systematic signal prioritization.
- **Verifier** : Cross-references associations against databases (e.g. FDA, medical literature) to identify "novel" signals absent from official reporting systems.
- **Proposer** : Generates testable hypotheses for novel associations by synthesizing biomedical literature, suggesting actionable causal explanations for clinical investigation.

Together, these agents allow **Echo** to transform unstructured patient discourse into structured pharmacovigilance signals, offering a new paradigm for early detection of safety concerns. In this paper, we systematically evaluate the components of **Echo** . Our contributions include: a proof-of-concept where we extract 640 drug–symptom associations from less than 200 Reddit posts, with Claude 3.5 Sonnet outperforming baselines as the **Explorer** agent ; identifying five associations absent from FDA safety data but strongly reported by patients, including pembrolizumab-linked daytime somnolence and nivolumab-associated pericardial effusion; identifying confounders shaping drug–symptom

relationships; and demonstrating through pre-2017 case studies that **Echo** might have anticipated later-recognized events, such as checkpoint inhibitor pneumonitis, months before regulatory updates. We also present an interactive user interface for exploring associations, evidence, confounders, and generated hypotheses. We conclude by reflecting on the ethical responsibility of amplifying patient voices in pharmacovigilance, and the limitations of relying on language models for each component.

## 2 Overview of Echo

Figure 1 presents the architecture of **Echo** , our multi-agent pharmacovigilance system designed to extract and analyze drug-symptom associations from patient discussions on social media. The system operates through four specialized agents, each instantiated by large language models and designed for a specific role in the pharmacovigilance pipeline. Further details about prompts and implementation of each agent can be found in our source code at `https://github.com/meghabyte/echo`.

The **Explorer** agent systematically collects patient discussions from cancer-related subreddits, identifying posts that mention oncology drugs and associated symptoms. For this paper, we focus on the same set of 187 Reddit posts scraped from a set of subreddits, including `r/cancer`, `r/lungcancer`, `r/leukemia`, etc. [1] These raw discussions are then processed by the **Analyzer** agent (e.g. Claude 3.5 Sonnet) which extracts structured drug-symptom associations and computes three quantitative metrics: temporal weight (proximity between drug administration and symptom onset), confidence (patient's expressed certainty), and community support (level of forum engagement). The three quantitative metrics are calculated as follows: temporal weight ranges from 0-1 based on explicit patient statements about timing between drug initiation and symptom onset; confidence weight (0-1) reflects the certainty language used by patients (e.g., "definitely caused by" vs. "might be related to"); and community support (0-1) measures engagement through upvotes and supportive replies. The ground truth score from the **Verifier** is set to $ln(FAERS_{count} + 1)$, where 0 indicates no recorded instances in the database. Finally, the **Proposer** agent generates mechanistic hypotheses for novel associations by synthesizing relevant biomedical literature, providing potential causal explanations that could guide further clinical investigation. As illustrated with the imatinib-parosmia case study, **Echo** can identify patient-reported side effects years before they appear in official safety databases, demonstrating the system's potential for early pharmacovigilance signal detection (Figure 1).

## 3 Experimental Results

We now present experimental results evaluating the different components of **Echo** .

### 3.1 Comparing Explorer Agents

Table 1: Performance comparison of different scraper agent approaches

| Agent for **Explorer** | Keyword Baseline | Claude 3.5 Haiku | Claude 3.5 Sonnet |
|---|---|---|---|
| Total Drugs | 40 | 141 | 131 |
| Total (Drug, ADR) Associations | 378 | 288 | 640 |
| Recall (Restricted Set) | 0.22 | 0.08 | 0.27 |

We first evaluated three different approaches for our **Explorer** agent to extract drug-adverse drug reaction (ADR) associations from Reddit discussions: **(1)** a keyword-based baseline that searches for exact drug and symptom term matches based on a standard list of 71 chemotherapy drugs and 78 side effects used in the FAERS dataset, **(2)** Claude 3.5 Haiku for lightweight natural language processing, and **(3)** Claude 3.5 Sonnet for more sophisticated contextual understanding. As shown in Table 1, while the keyword baseline identified fewer total drugs (40), it achieved competitive recall (0.22) on the restricted set of 20 drugs that were successfully identified by all three methods. Recall was evaluated over this restricted set using the **Verifier** agent against the top-5 highest frequency adverse effects for each drug in the FAERS database. While recall scores may appear low (0.08-0.27), this

---

[1] Our use of Reddit data is purely for research purposes, and since Reddit restricted API access in 2023 Reddit Inc. [2023], we identified posts using SerpApi, a web scraping service that provides programmatic access to search engine results. This limits the total number of posts we are able to access for this report.

reflects fundamental differences between patient-reported quality-of-life issues discussed online and the severe, often fatal events that dominate official databases (e.g., 'death,' 'off label use'). This gap underscores the complementary value of social media pharmacovigilance.

## 3.2 Novel Drug-ADR Associations

| Association | Temporal | Confidence | Community | Verifier | Patient Quote |
|---|---|---|---|---|---|
| Pembrolizumab → Daytime somnolence | 10/10 | 10/10 | 10/10 | 0.0 | asleep at my desk mid-afternoon |
| Paclitaxel → Laryngeal edema | 10/10 | 10/10 | 10/10 | 0.0 | my throat was closing up |
| Nivolumab → Pericardial effusion | 10/10 | 10/10 | 8/10 | 0.0 | two emergency pericardial windows |
| Fluorouracil → Abdominal bloating | 8/10 | 8/10 | 10/10 | 0.0 | she feels bloated especially after food |
| Oxaliplatin → Social withdrawal | 8/10 | 8/10 | 8/10 | 0.0 | I've withdrawn from all social interactions |

Table 2: Top 5 potentially novel drug-symptom associations flagged by the AI system.

Recall that the main goal with **Echo** is to identify *novel* associations between drugs and adverse reactions. Here we identify associations potentially underrepresented in official databases rather than definitively novel associations. Absence from FAERS may reflect underreporting, terminology differences, or true novelty—further clinical investigation is required to distinguish these cases. Table 2 shows associations with high patient-reported confidence but zero FAERS occurrences, warranting additional scrutiny rather than immediate acceptance as novel signals. The associations span diverse symptom categories, from neurological effects like daytime somnolence with pembrolizumab to serious cardiovascular complications like pericardial effusion with nivolumab, highlighting the potential for social media pharmacovigilance to capture underreported or emerging safety signals that warrant further clinical investigation.

## 3.3 Analyzing Confounding Variables with Analyzer

Recall that, unlike prior approaches leveraging social media to support pharmacovigilance, **Echo** can leverage large language models to capture more complex relationships than simple drug-reaction associations. This includes *confounding variables*, or potential explanations for the observed reaction that may not be explained by the drug alone. For example, in the associations listed in Table 2, confounding variables identified by **Echo** include a patient taking pembrolizumab also recovering from surgery, making daytime somnolence more likely, or a patient taking oxaliplatin, experiencing mood changes and social withdrawal, also dealing with the medical trauma of a cancer diagnosis.

| Confounding variable | Example (Drug, ADR) |
|---|---|
| surgical menopause | (letrozole, weight loss), (zoledronicacid, cognitive impairment), (tamoxifen, depression) |
| multiple myeloma | (daratumumab, nausea and vomiting), (dexamethasone, renal failure), (lenalidomide, general malaise) |
| recent surgery | (bevacizumab, impaired wound healing), (pegfilgrastim, sepsis), (pembrolizumab, pain) |
| male | (cabozantinib, hypertension), (oxaliplatin, social withdrawal), (ciltacabtageneautoleucel, CD4 lymphopenia) |
| young age | (bleomycin, alopecia), (etoposide, fatigue), (epcoritamab, serious infections) |

Table 3: Examples of salient confounding variables and their associated (drug, ADR) pairs.

For a more rigorous glance at confounding variables, we report in Table 3 the most salient confounding variables analyzed by **Analyzer** , where saliency is measured as the number of unique symptoms the confounding variables are associated with across different drugs. These results support intuition around the clinical relevance of patient characteristics and comorbid conditions in shaping observed drug–ADR associations, while also underscoring from a statistical perspective how such variables can

act as systematic sources of bias, attenuating or inflating pharmacovigilance signals if not properly accounted for. Looking ahead, recent work on causal inference from unstructured language data [Ke et al., 2024] motivates exploring whether associations identified as novel by Echo tend to be accompanied with more confounding variables, and whether patients are more likely to mention some confounders over others—an exciting direction for integrating text-based confounder discovery with pharmacovigilance.

## 3.4   Hypothesis Generation with Proposer

Finally, the Proposer agent demonstrates a key advancement over prior pharmacovigilance approaches by automatically generating mechanistic hypotheses for novel drug-symptom associations. As shown in Table 4, the system leverages large language models to connect patient-reported associations with relevant biomedical literature, producing plausible causal explanations that span diverse biological pathways—from immune-mediated sleep disruption with pembrolizumab to microbiome dysbiosis with fluorouracil. Notably, the Proposer agent autonomously identifies and cites relevant scientific references to support each hypothesis, eliminating the need for manual literature review that characterized earlier work [Ransohoff et al., 2018]. This forward-looking approach moves beyond retrospective validation to provide actionable mechanistic insights that could guide targeted clinical investigations, representing a significant methodological advance from prior systems that required manual labeling, were limited to specific ADR categories, and lacked the capability to systematically analyze temporal relationships, patient confidence, and potential confounding factors in drug-symptom associations.

| Association | Proposer Report |
|---|---|
| Pembrolizumab → Daytime somnolence | Primary Hypothesis: Immune-Mediated Sleep-Wake Cycle Disruption. Pembrolizumab's PD-1 checkpoint inhibition may trigger excessive immune activation that disrupts hypothalamic sleep-wake regulatory circuits[Krueger et al., 2016, Imeri and Opp, 2009]... |
| Paclitaxel → Laryngeal edema | Primary Hypothesis: Hypersensitivity-Mediated Laryngeal Inflammation. Paclitaxel is formulated with Cremophor EL (polyoxyethylated castor oil), a known histamine liberator that can cause immediate hypersensitivity reactions in 2–4% of patients [Weiss et al., 1990]... |
| Nivolumab → Pericardial effusion | Primary Hypothesis: Immune-Related Adverse Event (irAE) via T-cell Hyperactivation. Nivolumab's blockade of the PD-1/PD-L1 checkpoint pathway removes critical immune tolerance mechanisms, potentially triggering autoimmune-mediated pericarditis... |
| Fluorouracil → Abdominal bloating | Primary Hypothesis: Gastrointestinal Microbiome Disruption and Dysbiosis. Fluorouracil's cytotoxic mechanism through thymidylate synthase inhibition may significantly alter the intestinal microbiome composition. This microbial imbalance could result in increased intestinal gas production and impaired gas clearance, manifesting as abdominal bloating... |
| Oxaliplatin → Social withdrawal | Primary Hypothesis: Platinum-Induced Neuroinflammation and Limbic System Dysfunction. Oxaliplatin's platinum compounds may accumulate in brain regions critical for social cognition, particularly the prefrontal cortex and limbic system. |

Table 4: AI-generated mechanistic hypotheses for novel drug–symptom associations.

## 4   Case Studies

We now present case studies presented by Echo . These case studies demonstrate how Echo could help amplify patient-driven early warning of potential side-effects of treatment. Therefore, here we restrict the Explorer agent to Reddit posts before 2017, in order to validate their aid with early warning. Likewise, we restrict the Verifier agent to draw signal only from FAERS events and drug label information before 2017. We emphasize that systematic validation across larger datasets would be needed to establish reliable predictive performance.

### 4.1 Pneumonitis: (nivolumab, pneumonitis) and (pembrolizumab, pneumonitis)

Immune checkpoint inhibitors (ICIs) like pembrolizumab (Keytruda) and nivolumab (Opdivo) have transformed cancer treatment by blocking PD-1/PD-L1 interactions to enable immune recognition of tumor cells [Zhu et al., 2020, Spagnolo et al., 2022, Leroy et al., 2017]. However, these therapies can cause pneumonitis—lung inflammation that impairs gas exchange, causes respiratory distress, and in severe cases leads to treatment discontinuation or death [Wang et al., 2024]. While clinical trials report 1-5% incidence, very recent real-world studies suggest higher rates, especially in lung cancer patients [Spagnolo et al., 2022, Zhu et al., 2020].

Our **Explorer** agent identified multiple examples of patient experience with pneumonitis **before 2017** from Reddit forums, including:

- *"I developed pneumonitis and had to be hospitalized for 5 days, which is also why i haven't resumed treatment "* (nivolumab, r/cancer)
- *"My aunt finished chemo and keytruda but developed severe pneumonitis adverse event from keytruda..."* (pembrolizumab, r/cancer)
- *"I had a 3 week fever, it was pretty awful. But got through it..."* (nivolumab, r/cancer)

Furthermore, **Analyzer** highlighted these associations due to their high temporal (average of 1.00), confidence (average of 1.00), and community vote (average of 0.27) scores. Pre-2017 prescribing information for nivolumab report pneumonitis as a rare side-effect from clinical trials (around 0.4% in clinical trials), so it is likely that our **Verifier** might have had flagged this association as lacking strong official data [Food and Administration, 2015]. Indeed, in a later retrospective study published in 2018, researchers observed that pneumonitis from drugs, including pembrolizumab and nivolumab, for patients with lung cancer was far more likely than previously reported from clinical trials (19% vs. 2.7%) [Tay and Califano, 2018]. Subsequently, immunotherapy-induced pneumonitis has received considerable more attention from medical community. This case study shows patient discussions surfaced this toxicity before widespread clinical recognition.

### 4.2 Neuromuscular complications: (nivolumab, myasthenia gravis-like autoimmune neuropathy)

Immunotherapy can also trigger neuromuscular autoimmune syndromes, including conditions resembling **myasthenia gravis (MG)\*\***. In these cases, patients develop antibodies or immune responses that impair neuromuscular junction transmission, which can progress rapidly and may overlap with other immune-related neuromuscular toxicities such as myositis or peripheral neuropathies [Suzuki et al., 2017, Johansen et al., 2019]. Early recognition of MG is critical, since diagnosis is often difficult in oncology patients where fatigue and weakness may be attributed to disease burden or other treatments.

When applying **Echo** , the **Explorer** identified a patient experience from **2015** (*"horrible MG like autoimmune neuropathy after 2 months of tx. Unable to talk...previously very active and independent. "* (nivolumab, r/medicine)), which was emphasized with high temporal (1.0) and confidence (1.0) scores from the **Analyzer** . Prior to 2018, reports of immunotherapy-associated MG-like neuropathy were largely anecdotal and confined to isolated case reports, our **Verifier** provides a score of 0. Notably, the 2025 FDA label update for nivolumab (Opdivo) expanded the *Neuromuscular/Neurologic* adverse event section to explicitly include myasthenic syndrome and myasthenia gravis (including exacerbation), Guillain-Barré, and demyelinating conditions such as myelitis—broadening from its original 2016 label. This shift illustrates how regulatory labeling lags behind patient-level signal emergence [U.S. Food and Drug Administration, 2025b].

### 4.3 Hepatotoxicity: (regorafenib, hepatotoxicity)

A particularly adverse side-effect from cancer-fighting drugs is the hepatotoxicity associated with regorafenib, an oral multi-kinase inhibitor approved for treatment of metastatic colorectal cancer and gastrointestinal stromal tumors (GIST). It inhibits angiogenic, stromal, and oncogenic kinases (such as VEGFR, TIE2, PDGFR, FGFR, and others) and is used in patients who have progressed on prior standard therapies. Clinical trials (e.g. CORRECT for CRC, GRID for GIST) observed hepatotoxic effects: elevated liver enzymes (ALT, AST, bilirubin), instances of hepatic necrosis, and

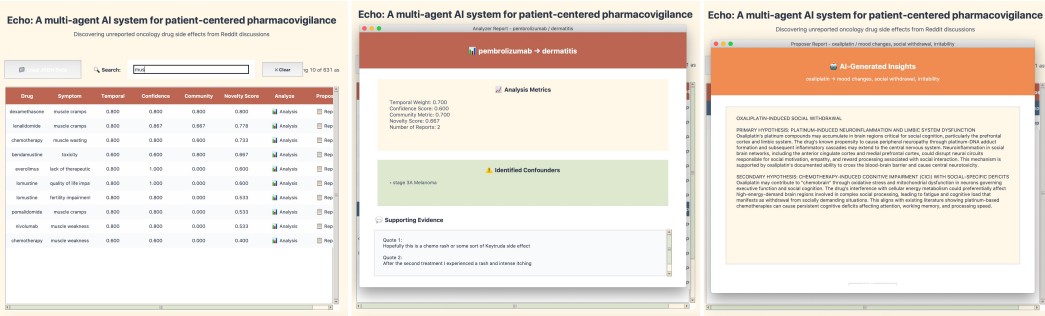

Figure 2: Overview of the user interface for **Echo** . Users can search through a database of (drug, side-effect) associations extracted by **Explorer** , including sorting via novelty score calculated by the **Verifier** (left). Clicking on a report from **Analyzer** reveals analysis scores, potential confounders for the associations (e.g. melanoma for dermatological symptoms), and supporting quotes from real-world patients (middle). Finally, proposals capture hypothesis for each association from **Proposer** to aid members of the medical community (right).

in some trials severe or fatal liver injury. The FDA prescribing information at initial approval (2012) already included a boxed warning for "severe and sometimes fatal hepatotoxicity." [U.S. Food and Drug Administration, 2012, Zhao and Zhao, 2017, Uetake et al., 2018].

**Echo** successfully identified the association between regorafenib and hepatotoxicity, with **Explorer** identifying the following patient examples **from 2015**:

- *"that's the medicine they put my dad on for Stage 4 Colon Cancer. They said it would slow the tumors... Instead, his liver shut down. He became extremely jaundice... check to see if her skin is turning ashy/yellow. "* (regorafenib, r/cancer)

- *"our latest update was that the medicine is doing nothing to deter the growth and is "overworking her liver""* (regorafenib, r/cancer)

**Analyzer** grouped these examples under the overall hepatotoxicity side-effect, and highlighted the (regorafenib, hepatotoxicity) association due to the high average community vote score (0.6). Because FDA labels as early as 2012 provide hepatotoxicity warnings, our **Verifier** would not mark this association as novel. Nevertheless, it was only until later that real-world data revealed that the incidence and severity of hepatotoxicity under regorafenib may be higher or more variable than clinical trial reports suggest. For instance, a 2017 meta-analysis across 14 trials (2,213 subjects) found substantial incidence of both all-grade and high-grade hepatotoxicity including elevated bilirubin and transaminases beyond what some individual trials had emphasized. Furthermore, the 2025 version of the FDA's label expands guidance on monitoring frequency, thresholds for dose interruption or discontinuation, and clearer definitions of severity.[U.S. Food and Drug Administration, 2025c]. Thus, **Echo** 's ability to detect patient-reported signals about regorafenib-linked hepatotoxicity in parallel with accumulating real-world evidence might have helped bridge the lag between early warning (from trials or scattered reports) and updated regulatory guidance.

## 5 User Interface

Finally, to facilitate exploration and analysis of the extracted drug-symptom associations, we developed an interactive visualization interface for **Echo** that enables researchers to efficiently examine the discovered pharmacovigilance signals (Figure 2). The interface presents a searchable tabular view where each row represents a unique drug-symptom pair, with columns displaying the averaged temporal weight, confidence score, and community metric calculated across all supporting evidence instances. Users can perform intelligent filtering by drug name or symptom description to rapidly locate associations of interest. For detailed analysis, the interface provides two complementary report mechanisms accessible through interactive buttons: an "Analyzer Report" that presents the underlying evidence from **Analyzer** , including identified confounders and supporting patient quotes extracted from Reddit discussions, and a "Proposer Report" that synthesizes AI-generated insights from **Pro-**

**poser** with hypothesis for the association. The interface processes the JSON-formatted output from **Echo** 's extraction pipeline and automatically aggregates metrics across multiple evidence instances, providing researchers with both granular access to individual patient reports and summary-level insights for prioritizing further investigation.

Sorting associations on the user interface by the Novelty score, provided by the **Verifier** , helps identify novel associations, such as those mentioned in Section 3.2. Overall, **Echo** can serve as a complementary early warning system for the pharmacovigilance community, bridging the critical gap between patient-reported experiences and regulatory safety updates to accelerate the identification of emerging adverse drug reactions before they become widespread clinical concerns.

# 6  Related Work

**Social media in medical research**    A substantial literature mines user-generated content to study health behaviors, conditions, and adverse drug reactions (ADRs). Early work established social media as a viable signal for pharmacovigilance and synthesized methods for ADR detection from noisy, informal text [Sarker and Gonzalez, 2015, Nikfarjam et al., 2015]. Reddit, with long-form, diagnosis- and treatment-focused threads, has been used to track health topics [Park et al., 2018], characterize oncology conversations [Thomas et al., 2019], and quantify ADR severity with signals derived from Reddit that correlate with FAERS outcomes [Lavertu et al., 2021]. Most recently, comparative analyses directly benchmark Reddit-mined adverse events against the FDA Adverse Event Reporting System (FAERS), highlighting complementary coverage and differences in event mix [Hayes and colleagues, 2024].

**Pharmacovigilance**    Conventional post-marketing safety relies on spontaneous reporting systems and large real-world data (RWD) networks. Disproportionality methods such as the Proportional Reporting Ratio (PRR) and empirical Bayes models (e.g., MGPS) underpin signal detection in FAERS [U.S. Food and Drug Administration, 2018, Almenoff et al., 2003, Zorych et al., 2013]. In parallel, FDA's Sentinel Initiative conducts active surveillance across claims and EHR data to generate real-world evidence at scale [U.S. Food and Drug Administration, 2024, 2019, 2025a]. These systems are rigorous but can lag emerging, rare, or subgroup-specific toxicities; complementary sources (e.g., patient forums) may narrow that gap by providing earlier, hypothesis-generating signals that can be triaged to formal pipelines.

**Data-driven early warning systems**    A broad line of work uses web and social data for public-health surveillance and early event detection [Fung et al., 2015, Paul and Dredze, 2014, Broniatowski et al., 2015]. Classical scan statistics remain foundational for detecting spatiotemporal anomalies in health data streams [Kulldorff et al., 2005]. At the same time, high-profile failures (e.g., Google Flu Trends) underscore risks of concept drift, confounding, and lack of ground truth when deploying web signals without rigorous validation and calibration [Lazer et al., 2014]. Our approach positions Reddit-derived associations as *early alerts* that must be reconciled with established pharmacoepidemiologic tools (e.g., disproportionality, cohort studies, Sentinel analyses), aligning with best practices learned from this literature.

# 7  Ethics and Limitations

Our approach raises important ethical and methodological considerations. First, Reddit posts reflect only what patients choose to share, which may not correspond to the full set of symptoms they experience or those a clinician would observe; this introduces reporting bias and limits the generalizability of the signals we detect. Second, Reddit imposes restrictions on data collection and use, and any pharmacovigilance system must remain compliant with platform policies while recognizing that our pipeline could be adapted to other online health forums where appropriate permissions are clearer. Third, even though we analyze publicly available content, users may not have intended their posts for secondary research in sensitive contexts such as oncology and adverse drug events. Respecting privacy norms, protecting anonymity, and acknowledging the potential for re-identification are therefore essential. These limitations highlight the need for caution: social media–derived signals should be treated as complementary, hypothesis-generating inputs rather than substitutes for clinical evidence, and they must be handled within ethical and regulatory frameworks that safeguard patient trust.

# 8 Usage of language models

The Claude Sonnet 3.5 language model from Anthropic was significantly used throughout this project, including for help with writing, code generation for experiments, and related works. Results were manually verified, and API keys for both language model API calls and SerpAPI to access Reddit posts were externally provided.

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
