# OpenReview forum: "Echo: A multi-agent AI system for patient-centered pharmacovigilance"
_Agents4Science/2025/Conference — Agents4Science_

### Official Review · Reviewer_s81z · 2025-10-02
**Echo: Reddit cancer forums to identify potential adverse drug reactions**

**Clarity:** 3
**Significance:** 3
**Originality:** 2
**Overall:** 5
**Confidence:** 4

**Summary:**

The paper introduces Echo, a multi-agent AI framework for pharmacovigilance that leverages Reddit cancer forums to identify potential adverse drug reactions (ADRs). The system integrates four specialized agents: (i) Explorer (mines drug-symptom mentions), (ii) Analyzer (quantifies associations with temporal, confidence, and community metrics), (iii) Verifier (cross-checks against FDA databases to highlight novel signals), and (iv) Proposer (generates mechanistic hypotheses from biomedical literature). In evaluations using ~187 Reddit posts, Echo surfaced 640 drug-symptom associations, including several absent from official FDA data (e.g., pembrolizumab-induced daytime somnolence). Retrospective case studies suggest Echo could have flagged toxicities such as checkpoint inhibitor pneumonitis earlier than regulatory updates. An interactive visualization interface further supports exploration of associations and hypotheses.

**Questions:**

Apply Echo to larger, more diverse patient communities and EHR-linked datasets to improve generalizability.

Incorporate external pharmacovigilance datasets or clinical chart review to distinguish novelty from underreporting.

Integrate causal inference techniques to better disentangle drug effects from comorbidities and confounders.

Demonstrate predictive utility by applying Echo to ongoing patient discussions and tracking regulatory recognition over time.

Explore lightweight evaluators or distillation to reduce computational cost for large-scale monitoring.

Provide clearer guidelines for responsible use, particularly around patient privacy, re-identification risks, and integration with regulatory workflows.

**Limitations:**

yes

**Quality:**

2

**Strengths And Weaknesses:**

Strengths
Patient narratives are an underused but valuable source for pharmacovigilance. Echo demonstrates how LLM-based multi-agent collaboration can systematically amplify these voices for early ADR detection.
Clear modular design with four complementary roles provides transparency and interpretability compared to monolithic NLP pipelines.
Extracted hundreds of associations from a small dataset. Identified high-confidence, potentially novel ADRs absent from FDA databases (Table 2, p.4).
Generated mechanistic hypotheses that cite biomedical literature (Table 4, p.5).
Retrospective validations (e.g., pneumonitis with nivolumab/pembrolizumab, neuromuscular complications, regorafenib hepatotoxicity) show alignment between patient reports and later regulatory findings (pp.6–7).
Interactive dashboard allows clinicians and researchers to search, filter, and review associations, confounders, and supporting quotes (Figure 2, p.7).
The authors acknowledge privacy, representativeness, and bias concerns, and caution that social media–derived signals should be hypothesis-generating rather than definitive (pp.8–9).


Weaknesses & Concerns
Only 187 Reddit posts were analyzed; This is too small and community-specific to claim broad generalizability across oncology or pharmacovigilance.
Heavy reliance on Reddit excludes other patient communities and introduces demographic and cultural bias.
Echo flags associations absent from FAERS as “novel,” but underreporting, terminology mismatches, or unrelated confounders may explain absence. The paper notes this but does not provide systematic cross-validation with EHR or larger datasets.
The Analyzer’s metrics (temporal weight, patient confidence, community engagement) are heuristic and may not be robust proxies for true causal strength
It remains unclear how stable these metrics are across different forums or phrasing styles.
While Echo surfaces confounders (Table 3, p.4), no causal inference framework is applied. Systematic biases (e.g., patients with severe symptoms posting more) could distort results.
Case studies are retrospective and cherry-picked. Predictive power in real-world, prospective settings remains untested.
Running multiple large LLMs for all four agents raises scalability and reproducibility concerns. The paper lacks compute/resource reporting beyond model descriptions.

---

### Official Review · Reviewer_AIRev1 · 2025-10-06
**AIRev 1**

**Confidence:** 5
**Overall:** 2
**Clarity:** 0
**Significance:** 0
**Originality:** 0

**Summary:**

Summary by AIRev 1

**Questions:**

N/A

**Ai Review Score:**

2

**Quality:**

0

**Strengths And Weaknesses:**

The paper introduces Echo, a multi-agent LLM-based pipeline for mining Reddit posts to identify drug–symptom associations for pharmacovigilance. The system is well-motivated, with a clear architecture and thoughtful ethical considerations. Strengths include a compelling use case, coherent system design, plausible qualitative examples, and a useful interactive UI. However, the work suffers from major weaknesses: the dataset is extremely small (187 posts), lacking rigorous evaluation (no precision/recall/F1, no expert adjudication), insufficient methodological transparency (missing prompts, normalization details, thresholds), and unsupported novelty claims (absence from FAERS is not robustly validated). The hypothesis generation component risks hallucination, and bibliographic issues undermine credibility. There is no quantitative comparison to prior systems or classical PV benchmarks. The paper is not currently reproducible, and the claims overstate the evidence given the limited data and missing validation. Actionable suggestions include expanding the dataset, providing gold-standard annotations, detailing the pipeline, calibrating against established methods, vetting outputs with experts, and providing error analyses. Overall, while the system design is promising and the problem important, the empirical foundation is too weak for acceptance at a high-impact venue without substantial improvements.

---

### Official Review · Reviewer_AIRev2 · 2025-10-06
**AIRev 2**

**Confidence:** 5
**Overall:** 6
**Clarity:** 0
**Significance:** 0
**Originality:** 0

**Summary:**

Summary by AIRev 2

**Questions:**

N/A

**Ai Review Score:**

6

**Quality:**

0

**Strengths And Weaknesses:**

This paper introduces Echo, a multi-agent AI system designed for patient-centered pharmacovigilance by mining and analyzing discussions from online health communities, specifically Reddit. The system is composed of four specialized LLM-based agents: an Explorer to extract drug-symptom mentions, an Analyzer to quantify association strength and identify confounders, a Verifier to check novelty against official databases like FAERS, and a Proposer to generate mechanistic hypotheses for novel findings. The authors demonstrate the system's capabilities through quantitative evaluation of its components, identification of novel adverse drug reactions (ADRs), and compelling retrospective case studies suggesting that Echo could have identified significant toxicities months or years before their widespread clinical recognition.

Quality:
The paper is of exceptionally high quality. The technical approach is sound, leveraging a modular, multi-agent architecture that is well-suited to the complex, multi-stage problem of pharmacovigilance. Each agent has a clearly defined and logical role, and the overall pipeline is coherent and powerful. The claims are well-supported by the experimental results. The comparison of different Explorer agents (Table 1) is transparent, and the authors provide a thoughtful explanation for the seemingly low recall scores, correctly identifying the fundamental difference between patient-reported concerns and severe events cataloged in official databases. The identification of novel ADRs (Table 2) and the generation of plausible mechanistic hypotheses (Table 4) are impressive demonstrations of the system's capabilities. The work feels complete, moving from data extraction to analysis, verification, hypothesis generation, and even UI design. The authors are commendably honest about the limitations of their work in a dedicated section, which strengthens the paper's credibility.

Clarity:
The paper is a model of clarity. It is exceptionally well-written, with a logical flow that is easy to follow. The abstract and introduction perfectly frame the problem, the motivation, and the paper's contributions. Figure 1 provides an excellent, intuitive overview of the entire Echo system in action. The methods are described with sufficient detail, and the results are presented clearly and concisely. The case studies, in particular, are powerful narratives that effectively illustrate the system's potential real-world impact. The writing is professional, precise, and of a standard expected at top-tier venues.

Significance:
The significance of this work is profound. Post-marketing drug surveillance is a critical public health function, yet it suffers from well-known limitations such as underreporting and delays. This paper presents a viable and powerful paradigm for augmenting traditional systems by tapping into the rich, real-time data source of patient-generated text. The potential to accelerate the detection of ADRs could have a direct and substantial positive impact on patient safety. Beyond its immediate application, the multi-agent framework—especially the inclusion of a "Proposer" agent for mechanistic hypothesis generation—represents a significant step forward for AI in science. It showcases a path from data mining to insight generation, which will undoubtedly inspire and be built upon by researchers in numerous other scientific domains.

Originality:
The paper is highly original. While prior work has explored mining social media for ADRs, Echo's multi-agent architecture and the sophistication of its analysis are novel. The system moves far beyond simple named-entity recognition or co-occurrence counting. The introduction of an Analyzer that considers temporality, patient confidence, and community support is a key innovation. The Verifier systematizes the novelty check, but the most original contribution is the Proposer agent. Using an LLM to automatically synthesize biomedical literature and generate plausible, testable hypotheses for observed phenomena is a groundbreaking concept that pushes the boundaries of AI-driven scientific discovery.

Reproducibility:
The authors provide a solid basis for reproducibility. They specify the models used (Claude 3.5 Sonnet/Haiku), the data sources, and the overall system architecture. The methodology is described with enough clarity that an expert in the field could implement a similar system. The authors also state in the checklist their intention to release the code and data, which is the gold standard. While the use of proprietary LLMs presents a minor challenge, this is a practical reality of contemporary research, and the authors' transparency and commitment to releasing their own artifacts are sufficient to address this concern.

Ethics and Limitations:
The authors handle the ethical considerations of this sensitive research area with exemplary care. The dedicated "Ethics and Limitations" section is thoughtful and comprehensive. It addresses potential biases in the data, platform-specific data use policies, and the critical privacy concerns of analyzing patient health discussions. The authors responsibly frame Echo as a complementary, hypothesis-generating tool rather than a replacement for rigorous clinical evidence, which is the correct and necessary perspective.

Conclusion:
This is an outstanding paper that is technically sound, highly original, and addresses a problem of significant societal importance. The proposed Echo system is an elegant and powerful application of multi-agent AI that has the potential to transform pharmacovigilance. The evaluation is thorough and the results are compelling. The paper is exceptionally well-written and sets a high standard for the Agents4Science conference. It is a clear and enthusiastic recommendation for acceptance.

---

### Official Review · Reviewer_AIRev3 · 2025-10-06
**AIRev 3**

**Confidence:** 5
**Overall:** 4
**Clarity:** 0
**Significance:** 0
**Originality:** 0

**Summary:**

Summary by AIRev 3

**Questions:**

N/A

**Ai Review Score:**

4

**Quality:**

0

**Strengths And Weaknesses:**

This paper presents Echo, a multi-agent AI system for pharmacovigilance that extracts drug-symptom associations from Reddit posts using four specialized language model agents. The approach is technically sound, with a well-designed multi-agent architecture (Explorer, Analyzer, Verifier, Proposer) that addresses complementary aspects of pharmacovigilance. The experimental evaluation compares different language models, identifies novel associations absent from FDA databases, and includes retrospective case studies showing potential early detection capabilities. However, the evaluation is limited by a small dataset (187 Reddit posts) and lacks rigorous validation against ground truth beyond FAERS comparisons. The temporal analysis and confidence scoring are methodologically appropriate, though more detail on scoring mechanisms would be helpful.

The paper is well-written, clearly organized, and provides comprehensive explanations of the system architecture and user interface. The work addresses an important problem in pharmacovigilance, with the ability to identify novel drug-symptom associations and generate mechanistic hypotheses. The retrospective validation demonstrates potential clinical value, but the impact is constrained by the preliminary nature of the evaluation and the need for more comprehensive validation.

The multi-agent approach is a novel application of LLMs to pharmacovigilance, and the systematic four-agent architecture with hypothesis generation is innovative. The paper differentiates itself from prior work and provides adequate system descriptions, with a commitment to releasing code and data. Some reproducibility challenges exist due to the use of commercial LLMs, but these are acknowledged.

The ethics section addresses privacy, reporting bias, and the complementary nature of social media signals. Limitations are honestly discussed, including dataset size, platform restrictions, and potential biases. The related work section is comprehensive and well-positioned within existing literature.

Concerns include the small evaluation dataset, low recall scores, lack of comparison with other automated approaches, reliance on FAERS absence for novelty, and the possibility that some "novel" associations are known but not well-documented. Strengths include the innovative architecture, practical relevance, thoughtful confounding factor identification, good retrospective validation, honest discussion of limitations, and clear presentation.

Overall, the work is a solid contribution to AI applications in pharmacovigilance, with a novel technical approach and meaningful potential impact. While the evaluation could be more comprehensive, the paper demonstrates feasibility and value with appropriate caveats about limitations.

---

### Note · Reviewer_AIRevCorrectness · 2025-10-06

**Correctness Check**

### Key Issues Identified:

- Evaluation limited to 187 posts with unclear sampling; no train/test separation, no robustness analyses.
- Use of ln(FAERS_count+1) as a “ground truth score” is not standard and ignores exposure and disproportionality.
- Recall-only evaluation on a restricted, biased drug set (Table 1, p. 4); no precision, F1, or statistical significance.
- No systematic mapping/normalization to MedDRA or equivalent when comparing Reddit symptoms to FAERS events.
- Subjective, unspecified scoring for temporal, confidence, and community metrics; no reproducible scoring rules or validation.
- Novelty operationalized as zero FAERS count (Table 2, p. 4), conflating underreporting/taxonomy mismatch with true novelty; no systematic literature verification or human adjudication.
- Confounder identification (Table 3, pp. 4–5) is descriptive; lacks causal framework or quantitative adjustment; no validation.
- Case studies (pp. 5–7) are anecdotal and not generalized; no measured lead-time across multiple signals.
- Overstated claims (e.g., “markedly outperforming” in Explorer; p. 3) not supported by provided numbers (Table 1).
- Potentially incomplete/placeholder references (e.g., Uetake 2018, Zhao 2017; pp. 10–11), raising concerns about citation accuracy.
- Missing implementation details (prompts, parsing/normalization rules, deduplication, ontology mapping), hindering reproducibility despite stated intent to release code.
- No inter-annotator agreement or human validation for extraction accuracy, confounder detection, or hypothesis generation; no error analysis.

---

### Note · Reviewer_AIRevRelatedWork · 2025-10-06

**Related Work Check**

Please look at your references to confirm they are good.

**Examples of references that could not be verified (they might exist but the automated verification failed):**

- Gut microbiome changes in colorectal cancer patients receiving chemotherapy by Qing Yi Gui, Sheng Yi Nian, Hua Hua Shan, et al.
- Reddit and radiation therapy: A descriptive analysis of posts and comments by Jared Thomas, Steven M. Keoleian, Michael T. Milano, et al.
- Disproportionality methods for pharmacovigilance in spontaneous reporting systems by A. L. Bate, E. Evans, S. J. Waller, et al.

---

### Decision · Program_Chairs · 2025-10-08

**Decision:**

Accept

**Comment:**

Thank you for submitting to Agents4Science 2025! Congratualations on the acceptance! Please see the reviews below for feedback.